# Economic Costs of Residential Fires: A Systematic Review

**Fahmida Saadia Rahman [1],\*, Wadad Kathy Tannous [1] , Gulay Avsar [1] , Kingsley Emwinyore Agho [2] , Nargess Ghassempour [1] and Lara A. Harvey [3]**

[1] School of Business, Western Sydney University, Parramatta, NSW 2150, Australia; k.tannous@westernsydney.edu.au (W.K.T.); g.avsar@westernsydney.edu.au (G.A.); n.ghassempour@westernsydney.edu.au (N.G.)

[2] School of Health Sciences, Western Sydney University, Penrith, NSW 2751, Australia; k.agho@westernsydney.edu.au

[3] Falls, Balance and Injury Research Centre, Neuroscience Research Australia, Randwick, NSW 2031, Australia; l.harvey@neura.edu.au

\* Correspondence: f.rahman@westernsydney.edu.au

**Abstract:** Globally, most fire-related deaths and injuries occur in residential areas. The aim of this systematic review is to report on the economic costs of residential fires from a societal perspective. Five databases (MEDLINE, EMBASE, EconLit, CINAHL, and Scopus) and grey literature were searched to identify studies that report economic or societal costs of residential fires with data from 1978 to 2021. There were no restrictions on study design. A narrative synthesis was undertaken based on the societal and economic costs reported for each included study. Seven studies from the United States, Canada, Australia, and Kuwait reported costs of residential fires. The costs of injuries and deaths were between USD 12 million and USD 5 billion, and between USD 75 million and USD 26 billion, respectively. The costs of treatment ranged from USD 0.3 million to USD 551 million, lost productivity from USD 12 million to USD 4 billion, and property damage from USD 8 million to USD 10 billion. This systematic review provides the most comprehensive evidence to date on the economic costs of residential fires. This study would offer insights into the effects of residential fires on diverse economic agents and aid in community fire prevention messaging and incentives.

**Keywords:** economic costs; fire; residential; injury; death





## 1. Introduction

Fire is one of the leading causes of death and disability worldwide [1]. Fire-related burns are the fourth most common source of unintentional trauma and have been categorized as a global priority for prevention [1]. Each year, there are an estimated 300,000 deaths due to fires and millions more people are left with lifelong injuries, disabilities, and disfiguring scars, often with resulting social isolation and economic losses for burn survivors and their families [2]. In low-, middle-, and high-income [3] countries, most fire-related deaths and injuries occur in residential properties [4–6].

Low- and middle-income countries (LMICs) experience over 90% of all fire-related deaths [1]. They are commonly considered more vulnerable to fire-related injuries and fatalities [2]. This is due, at least in part, to limited building and fire safety codes [7]. Moreover, fire safety measures are not ensured during the design, construction, and maintenance phases [7,8]. Despite this increased burden, there is limited research on LMICs on rates of injury/death and property damage resulting from residential fires. Studies conducted on LMICs reveal that fire is a regular problem in homes, workplaces, hospitals, and public places, and has become a serious threat, particularly for urban settlement [7,9–16]. However, most of these studies describe fire incidence and prevalence across these settings without the provision of information specifically on residential fires.

For high-income countries, fire incidents are also a major concern [17]. There are many studies that provide statistics on fire incidents, related injuries and/or deaths, and

property loss. Of these, limited studies report on residential fires. According to the National Fire Protection Association (NFPA), 27% of all fires in the USA in 2020 occurred in the home, resulting in 78% of all fire injuries, 75% of all fire deaths, and 40% of all direct property losses [18]. These statistics are an indication of the severity of the consequences of residential fires. Between 2005 and 2015, a total of 145,252 residential fires were reported in four provinces in Canada, causing casualties, 86% of which being injuries and 14% being deaths related to the residential fires [19]. Similarly, in England in 2021, around 83% of all primary fires were residential fires, resulting in 77% of all non-fatal casualties and 78% of all fire-related fatalities [20]. The proportion of residential fire-related deaths is also high in Sweden. Approximately, 100 fire deaths per year (more than one death per week) were reported in Sweden [21,22], and residential fires account for three quarters of those fire-related fatalities [23]. Similarly, fire is considered a common hazard to people, property, and the environment in Australia [24]. In 2020, fires resulted in 122 deaths and an estimated 3700 hospitalisations due to fire-related injuries [25].

Residential fire statistics and associated injuries and/or deaths are based on reported incidents to the fire brigades. Recent studies have revealed that the number of residential fire incidents is higher than the reported figures. This is often due to residents not reporting house fires, which may self-extinguish or be extinguished with the assistance of residents or bystanders, among other reasons [26–28]. Therefore, the number of official residential fire incidents is lower than the true figures. Residential fires have generally been described in terms of the property damage and/or the number of related injuries and deaths and/or the number of fire brigades and fire trucks attended at the scene. The impact and associated costs of residential fires extend beyond that. They involve many stakeholders including individuals, response agencies, health services, other businesses, and governments.

The estimation of the economic cost [29] of residential fires has been under researched to date internationally. The aim of this study is to systematically review and report on the economic costs of residential fires from a societal perspective. This includes the costs of different stakeholders associated with residential fire incidents. This study would provide insights into the effects of residential fires on diverse economic agents and aid in community fire prevention messaging and incentives.

## 2. Materials and Methods

The Population/problem/phenomenon, Exposure, and Outcomes (PEO) framework [30] was used to formulate the systematic review question, guide the literature search strategy, identify relevant studies for review, and specify inclusion and exclusion criteria. Population included all populations, response agencies, and businesses who have experienced a residential fire and/or fire-related injury and/or death. Residential fire was the Exposure, and Outcomes included the costs of residential fires in terms of lost productivity and quality of life for individuals, and the costs of response agencies, the health service system, other businesses, and governments. The systematic review protocol was registered with the International Prospective Register of Systematic Reviews (PROSPERO) database (reference CRD42021222797).

### 2.1. Search Strategy

The search strategy (provided in Appendix A, Table A1) was developed in collaboration with input from an experienced information specialist (university librarian) and designed to be as extensive as possible to identify all eligible studies. Five databases (MEDLINE, EMBASE, EconLit, CINAHL, and Scopus) were searched using a variety of subheadings and free text terms. An iterative search strategy was performed with the key words residential, fire, injury, death, and cost, for instance, residen* or hous* AND fire* or smoke* or flame* AND injur* or burn* or death* or fatal* AND cost* or burden*.

The search on the databases was conducted with publication dates from 1 January 1978 to 31 December 2020 as per PROSPERO registration. The starting date of 1978 was chosen as it coincides with the date of the first introduction of legislation requiring the

installation of smoke alarms anywhere in the world (Montgomery County, MD, USA) [31]. This safety measure had significant effects on the impacts of residential fires due to its early warning system. Prior to publication, the search was updated until 31 December 2021 to capture the most recent eligible studies. However, of the new studies identified for 2021 and incorporated in a PRISMA diagram [32] in Figure 1, none were included in the final review. Reference lists of included articles were checked to identify any additional studies. In addition, the review included any relevant officially published Government/National reports from grey literature. For studies where the relevant study characteristic or outcome data are missing, the reviewers contacted the study authors.

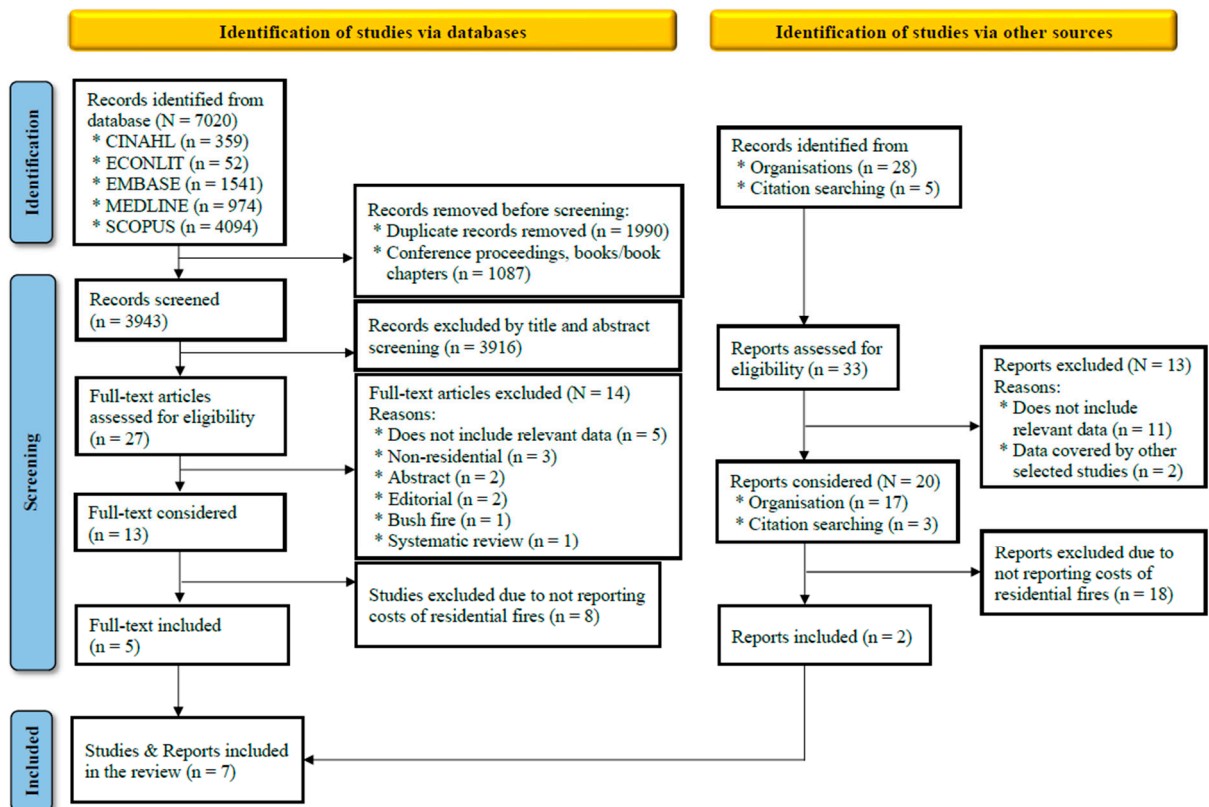

**Figure 1.** Preferred Reporting Items for Systematic Reviews and Meta-Analysis (PRISMA) flow diagram.

### 2.2. Eligibility Criteria

The eligibility criteria established for including and excluding studies are described in Table 1 based on the research question.

Conference proceedings, that are a collection of abstracts and papers presented at conferences, were excluded as they usually provide insufficient detail, and may not be peer-reviewed. Book chapters per se were excluded as they predominantly used secondary data; however, the review included the primary source of any secondary data/information identified in any of the book chapters.

### 2.3. Study Selection

Each database was searched, specifically tailoring the search syntaxes to each database. Endnote software (EndNote X9, Philadelphia, 2013) was used to manage the search results. The selection of studies for inclusion in the review was a three-step process (Figure 1) managed in Covidence (Covidence systematic review software, Veritas Health Innovation, Melbourne, Australia. Available at www.covidence.org (accessed on 14 April 2021)). In Step 1, two reviewers (FR and NG) independently screened study titles and abstracts to identify all potentially eligible studies meeting the inclusion criteria. In Step 2, two reviewers

(FR and LH) independently assessed the full text of all the identified potentially eligible studies to determine which studies would be considered in the review. In Step 3, the same reviewers (FR and LH) independently assessed whether the considered studies in Step 2 reported costs of residential fires. Discrepancies were resolved through discussion with a third reviewer (LH or KT) in order to reach consensus in each step.

**Table 1.** Eligibility criteria for the studies to be selected.

| | |
|---|---|
| Inclusion criteria | All articles and official reports published in peer-reviewed journals and in grey literature addressing residential fires with a focus on economic or societal cost/burden of residential fires incorporating the following:<br>■ Cost/burden of residential fire incidents and associated injuries/deaths.<br>■ Cost/burden to individual, bystanders, and carers (in terms of lost productivity, income, and quality of life).<br>■ Costs of health service.<br>■ Costs of response agencies (fire service, ambulance, police, and other emergency services) to fires.<br>■ Cost/burden of business agents (the utility providers, insurance companies, and the loss of productivity/outcome of the employers).<br>■ Loss of residential structural property.<br>■ Cost/burden of the government to fires (in terms of the lost tax revenue, the cost of welfare payment, and funding/support to different response agencies and to the society). |
| Exclusion criteria | The review excludes studies:<br>■ Written before January 1978 and after December 2021.<br>■ On non-residential fires.<br>■ Published as editorials, abstracts, conference proceedings, and book chapters. |

*2.4. Data Extraction*

Data extraction of each selected study was conducted independently by three reviewers (FR, GA, and KT) using a standardized pilot-tested data extraction form developed and managed in Research Electronic Data Capture (REDCap) (electronic data capture tools hosted at Western Sydney University [33,34]). The data extraction form included the following information: lead author, title of study, type and year of publication, type and duration of data, study area, study design, aim/objective, study population and age range, cost information of residential fire incidents and associated injury/death, cost information of individuals, response agencies, health service, other businesses, and government.

The types of publication were grouped into two options: (i) peer-reviewed journal, and (ii) non-peer reviewed studies. Non-peer reviewed studies included government/non-government organisational, and national statistics, and reports, for instance, annual report, or commissioned report. The types of data were categorised into four options: (i) administrative data, (ii) clinical data, (iii) survey data, and (iv) secondary data. The study design was of three types: (i) randomised controlled trial (RCT), (ii) observational, and (iii) others. Observational studies included cohort studies, cross-sectional studies, case–control studies, and national statistics.

*2.5. Data Analysis and Synthesis*

All results were subject to double data entry by three independent reviewers (FR, GA, and KT). The studies were grouped by country for reporting. Due to the very limited data identified and the high diversity of the data, a quantitative synthesis and meta-analysis were not possible. Therefore, a narrative synthesis was undertaken based on the societal and economic costs for each identified study. Only a few studies reported on their population figure. Hence, the population figures, in instances where the studies did not include the number, were obtained from the official and unofficial statistical sources for the purpose of analysis. The review extrapolated the population for the years coinciding with the data collection period of the corresponding studies. The cost figures were converted to per annum, and expressed in US dollars (USD) in terms of both nominal values and constant

2022 US dollar, using the consumer price index (CPI) provided by the U.S. Bureau of Labour Statistics [35].

*2.6. Quality Assessment*

Two reviewers (FR and LH) independently assessed the quality of the included studies using a customised quality assessment tool (Table A2 in Appendix A). As this review includes a wide range of study designs and is not aimed at testing an intervention, none of the commonly used risk of bias (ROB) tools identified in the literature, such as (i) Critical Appraisal Skills Programme (CASP) [36], (ii) Scottish Intercollegiate Guidelines Network (SIGN) [37], (iii) National Institute for Health and Care Excellence (NICE) [38], (iv) Joanna Briggs Institute (JBI) [39], or (v) Newcastle-Ottawa Scale (NOS) [40], were deemed appropriate for the broad scope of our review. The commonly used tools were designed predominantly to assess the quality of the intervention studies. Therefore, a customised quality assessment tool was developed following the approach recommended by Wang et al. [41]. The nine domains common to most ROB tools, namely, selection, exposure, outcome assessment, confounding, lost to follow-up, analysis, selective reporting, conflict of interest, and other, were considered. Our customised critical appraisal tool included five items covering the study area and population, exposure and outcome (cost) assessment, reporting quality, and generalisability. There was a maximum score of five available for the studies. The quality of the studies was rated as very good (if scored 5), good (if scored 3–4), poor (if scored 1–2), and very poor (if scored 0). Any discrepancy or uncertainty in quality assessment was resolved by discussion.

## 3. Results

*3.1. Literature Search and PRISMA Chart*

The electronic database searches yielded a total of 7020 records. After identifying and removing duplicates, conference proceedings, and books/book chapters, 3943 records were screened by title and abstract with 27 articles determined to be eligible for full-text assessment in Step 1. Of these, 13 articles were considered for further full text-assessment as they reported on residential fires, and the remaining 14 were excluded in Step 2 for the reasons articulated in Figure 1. Of the 13 articles reporting on residential fire incidents, 5 met the eligibility criteria and 8 were excluded for not reporting any cost of residential fires (Figure 1).

An additional 33 records identified by hand-searching from other sources (i.e., grey literature and reference list of included studies) were assessed for eligibility in Step 1. Of these, 20 reports were considered for further assessment due to reporting on residential fires, and 13 were excluded in Step 2. Of the 20 reports, 2 were included as per the eligibility criteria and 18 were excluded for not reporting costs of residential fires (Figure 1).

A total of seven studies were finally included in the systematic review (Figure 1). A list of excluded articles that failed to meet the eligibility criteria in Step 2 is provided in Appendix A (Table A3). The findings from the articles that addressed residential fires but were excluded from the review in Step 3 due to not reporting on costs of residential fires are presented in Table A4 in Appendix A.

*3.2. Study Characteristics*

The characteristics of the seven included studies are presented in Table 2. Four of the studies were conducted in the United States [18,42–44], and one each in Canada [45], Australia [46], and Kuwait [47]. The review found five peer-reviewed journal articles and two organisational reports. All seven studies were observational, including cohort and cross-sectional studies. The selected studies used administrative data, survey data, and clinical data (Table 2).

**Table 2.** The characteristics of the included studies.

| Author and Year of Publication | Country | Study Population | Study Design | Data Collection Period | Type of Data | Type of Publication |
|---|---|---|---|---|---|---|
| McLoughlin, 1990 [42] | USA | Children aged between 0 and 19 years | Observational | 1985 | Administrative and survey data | Peer-reviewed |
| Yellman, 2018 [43] | Dallas, Texas, USA | 57,140 residents participating in a community-based smoke alarm installation program | Observational | 2006–2012 | Administrative data | Peer-reviewed |
| Lawrence, 2009 [44] | USA | At the population level | Observational | 1995–2003 | Administrative and survey data | Organizational report |
| Ahrens and Evarts, 2021 [18] | USA | At the population level | Observational | 2020 | Survey data | Organizational report |
| Banfield, 2015 [45] | Ontario, Canada | Fire injured adults (16 years and above) admitted to the provincial burn centre | Observational | 1995–2012 | Administrative and clinical data | Peer-reviewed |
| Tannous, 2018 [46] | NSW, Australia | 18 years and over | Observational | 2014 | Survey data | Peer-reviewed |
| Koushki, 2000 [47] | Kuwait | At the population level | Observational | 1994–1995 | Administrative and survey data | Peer-reviewed |

*3.3. Societal and Economic Cost of Residential Fire*

The estimated costs of residential fires and associated injuries and deaths are presented in Table 3. The costs per annum are described in 2022 constant USD. The overall costs of residential fires were almost USD 13 million in Dallas, USA [43], USD 242 million in Ontario, Canada, and in New South Wales (NSW), Australia, the corresponding amount was USD 435 million [46].

Three studies [42–44] reported the costs of injuries and deaths associated with residential fires in the USA. McLoughlin and McGuire [42] reported that 84.3% of fire and burn deaths in children happened due to a house fire, and the related cost of deaths was USD 8 billion. In a study conducted in Dallas evaluating a smoke alarm installation program, Yellman et al. [43] estimated the cost of injuries at USD 12 million. The third study, conducted at the population level by Lawrence et al. [44], estimated the costs of injuries and deaths to be USD 5 billion and USD 26 billion, respectively. Apart from the USA, the costs of injuries and deaths related to residential fires were reported in one study from Australia. In their study conducted in NSW, the most populous state in Australia, Tannous et al. [46] measured the cost of injuries to be USD 150 million and that of deaths to be USD 75 million (Table 3).

Only studies conducted in the USA [42–44] estimated the lost productivity/income with a reported range between USD 12 million and USD 4 billion. Two studies costed the lost quality of life, and this was measured to be worth USD 27 billion in the USA [44] and USD 61 million in NSW, Australia [46] (Table 3).

The costs of health service associated with residential fires were reported in the USA, Canada, and Australia. Yellman et al. [43] measured a treatment cost of around USD 0.3 million in their study conducted in Dallas, USA. Lawrence et al. [44] estimated the treatment cost (USD 551 million) including costs of emergency department (around USD 32 million), and hospital inpatient/outpatient care (USD 464 million) in the USA. The costs of treatment and hospital inpatient/outpatient care were USD 6 million and USD 7 million, respectively, in Ontario, Canada [45]. In their study, Banfield et al. [45] further quantified the costs of ambulance (almost USD 14,000) in Ontario, Canada. In NSW, Australia, the cost of treatment was reported at USD 7 million [46] (Table 3).

**Table 3.** The reported outcomes from the included studies.

| Author and Year of Publication | Extrapolated Population (for the Purpose of Analysis) | Number of Residential Fires, Associated Injuries and Deaths | Reported Costs of Residential Fires, Associated Injuries and Deaths (in Nominal US Dollar) | Reported Costs of Residential Fires, Associated Injuries and Deaths (in 2022 Constant US Dollar) | Other Reported Costs Associated with Residential Fires (in Nominal US Dollar) | Other Reported Costs Associated with Residential Fires (in 2022 Constant US Dollar) |
|---|---|---|---|---|---|---|
| McLoughlin [a], 1990 [42] | 70,261,000 [48] children aged between 0 and 19 years in 1985 | Deaths: 1231 [1.8 per 100,000 population] | Costs of deaths (USD 3.1 b [!]) | Costs of deaths (USD 8.4 b) | Lost income/productivity (USD 328 m [†]) | Lost income/productivity (USD 892.2 m) |
| Yellman [b], 2018 [43] | - | - | Costs of fires (USD 9.7 m), injuries (USD 8.6 m) annually | Costs of fires (USD 13.4 m), injuries (USD 11.9 m) annually | Lost income/productivity (USD 8.3 m), costs of health service (treatment—USD 0.2 m) annually | Lost income/productivity (USD 11.5 m), costs of health service (treatment—USD 0.3 m) annually |
| Lawrence, 2009 [44] | 305,694,910 [49] in 2008 | Injuries: 57,528 [18.8 per 100,000 population] annually; deaths: 3062 [1.0 per 100,000 population] annually | Costs of injuries (USD 3.1 b), deaths (USD 15.4 b) annually | Costs of injuries (USD 5.3 b), deaths (USD 26.3 b) annually | Lost income/productivity (USD 2.6 b) and quality of life (USD 15.6 b), costs of health service (treatment—USD 322 m, emergency department—USD 18.4 m, and hospital in-/outpatient care—USD 271.3 m) annually | Lost income/productivity (USD 4.4 b) and quality of life (USD 26.7 b), costs of health service (treatment—USD 550.6 m, emergency department—USD 31.5 m, and hospital in-/outpatient care—USD 464 m) annually |
| Ahrens and Evarts, 2021 [18] | 335,942,003 [50] in 2020 | Fires: 379,500; injuries: 11,900 [3.5 per 100,000 population]; deaths: 2630 [0.8 per 100,000 population] | - | - | Costs of property damage (USD 8.7 b) | Costs of property damage (USD 9.9 b) |
| Banfield [c], 2015 [45] | 4 million households in 2011 census (given in their study) | Fires: 6452 annually; injuries: 67 [1.7 per 100,000 population] annually; deaths: 92 [2.3 per 100,000 population] annually | Costs of fires (USD 192 m) annually | Costs of fires (USD 241.9 m) annually | Costs of health service (treatment—USD 5.1 m, and hospital in-/outpatient care—USD 5.3 m), ambulance (USD 10,847), property damage (USD 187 m) annually | Costs of health service (treatment—USD 6.4 m, and hospital in-/outpatient care—USD 6.7 m), ambulance (USD 13,667), property damage (USD 235.6 m) annually |

**Table 3.** *Cont.*

| Author and Year of Publication | Extrapolated Population (for the Purpose of Analysis) | Number of Residential Fires, Associated Injuries and Deaths | Reported Costs of Residential Fires, Associated Injuries and Deaths (in Nominal US Dollar) | Reported Costs of Residential Fires, Associated Injuries and Deaths (in 2022 Constant US Dollar) | Other Reported Costs Associated with Residential Fires (in Nominal US Dollar) | Other Reported Costs Associated with Residential Fires (in 2022 Constant US Dollar) |
|---|---|---|---|---|---|---|
| Tannous [d], 2018 [46] | - | - | Costs of fires (USD 369 m), injuries (USD 127.4 m), deaths (USD 63.7 m) | Costs of fires (USD 435.4 m), injuries (USD 150.3 m), deaths (USD 75.2 m) | Lost quality of life (USD 51.6 m), costs of health service (treatment—USD 5.5 m), fire service (USD 3.8 m), police and emergency service (USD 1.9 m), insurance companies (USD 115 m), individual out of pocket insurance cost (USD 5.2 m) | Lost quality of life (USD 61 m), costs of health service (treatment—USD 6.5 m), fire service (USD 4.5 m), police and emergency service (USD 2.2 m), insurance companies (USD 136 m), individual out of pocket insurance cost (USD 6.1 m) |
| Koushki [e], 2000 [47] | 1,895,000 [51] in 1994–1995 | Fires: 486; injuries: 113 [6.0 per 100,000 population]; deaths: 8 [0.4 per 100,000 population] | - | - | Costs of property damage (USD 4.1 m) | Costs of property damage (USD 7.5 m) |

[!] billion. [†] million. [a] Number of deaths, cost of deaths (by taking the median value for cost of deaths reported in the paper), lost productivity and life years for deaths in the table were calculated as per the proportion of deaths (84.3%) mentioned due to house fires in the study. [b] This review reported the cost figures including both the recipient and non-recipient control groups. [c] Costs in the table were calculated in terms of the US dollar using the currency exchange rate given in the study. [d] Costs in the table were calculated in terms of the US dollar using purchasing power parity (PPP) from the OECD at https://data.oecd.org/conversion/purchasing-power-paritiesppp.htm?fbclid=IwAR26duaP4UjuXHHgGg8VJUITnrqeAai_xdJlMyDJ1r4qG2W3EJ-5z8g0fec (accessed on 19 August 2022). [e] This review estimated that 68.6% of all fires occurred in residential houses using the given data in the study. Number of residential fires, related injuries and deaths, and property damage in the table were calculated as per the proportion of house fires (68.6%) estimated from the study.

Only Tannous et al. [46] reported the cost of fire service (USD 5 million), police and emergency service (USD 2 million), insurance companies (USD 136 million), and individual out-of-pocket insurance cost (USD 6 million) for NSW, Australia. Lost productivity/income, costs of health service (including ambulance, emergency department, and hospital inpatient/outpatient care), and property damage were not measured in Australia. The cost of property damage was reported in the USA (USD 10 billion) [18], in Ontario, Canada (USD 236 million) [45], and in Kuwait (USD 8 million) [47] (Table 3).

### 3.4. Quality Assessment

The quality of seven studies was assessed using the customised quality assessment tool (Table A2 in Appendix A). This review found three very good-quality studies and the rest four were good-quality studies. This review did not find any poor-quality study. The quality assessment score for individual studies is described in Table 4.

**Table 4.** Quality assessment score of the included studies.

| Author and Year of Publication | Domain and Questions * | | | | | Total Score |
| --- | --- | --- | --- | --- | --- | --- |
| | **Selection** | **Exposure** | **Outcome Assessment** | **Selective Reporting** | **Generalisability** | |
| | (Y = 1; N = 0) | (Y = 1; N = 0) | (Y = 1; N = 0) | (Y = 1; N = 0) | (Y = 1; N = 0) | |
| Mcloughlin, 1990 [42] | 1 | 1 | 1 | 1 | 0 | 4 |
| Yellman, 2018 [43] | 1 | 1 | 1 | 1 | 1 | 5 |
| Lawrance, 2009 [44] | 1 | 1 | 1 | 1 | 1 | 5 |
| Ahrens and Evarts, 2021 [18] | 1 | 1 | 1 | 0 | 1 | 4 |
| Banfield, 2015 [45] | 1 | 1 | 1 | 1 | 1 | 5 |
| Tannous, 2018 [46] | 1 | 0 | 1 | 1 | 1 | 4 |
| Koushki, 2000 [47] | 1 | 1 | 1 | 0 | 1 | 4 |

* Questions: Selection—Is the source population or source area well described? Exposure—Was the exposure accurately measured to minimise bias? Outcome assessment—Were the outcomes clearly defined? Selective reporting—Were the measured outcomes reported? Generalisability—Can the outcomes be applied to the general population?

## 4. Discussion

This is the first systematic review on the economic costs of residential fires and associated consequences on society, beyond just reporting the number of residential fire incidents, injuries, and deaths. Whilst many studies reporting fire statistics were identified by the search strategy, the majority were not specific to residential fires and could not be included as per the eligibility criteria. Further, the extracted data from the selected studies were very diverse; most studies did not include the size of their study population, and those which did include denominator data had very different population size, cohort, and reporting criteria. Studies reported costs of residential fire incidents along with associated injuries and/or deaths based on a population cohort of children only [42] or of adults only [45,46] or of houses involved in a smoke alarm installation program [43]. This review was limited in the undertaking of any quantitative analysis including a meta-analysis or subgroup analysis due to the limited number of studies included and the wide range of data extracted from those included studies. In addition, per capita costing, or costs of per 100,000 population of incidents were unable to be determined due to the missing data in the papers.

Only seven studies were selected that had met all the eligibility criteria of this review. They were all conducted on high-income countries (the USA, Canada, Australia, and Kuwait). Not a single study was identified that had examined the economic costs of residential fires in LMICs. The identified studies reported on the number of residential fire

incidents, the nature of those incidents, and associated injuries or deaths. Articles that were searched or read but did not contain any measure of economic costing were consequently not included due to the eligibility criterion. This review found that the costs of injuries and deaths in the seven studies were between USD 12 million and USD 5 billion, and between USD 75 million and USD 26 billion, respectively. Lost income/productivity costs ranged from USD 12 million to USD 4 billion, lost quality of life costs between USD 61 million and USD 27 billion, costs of treatment from USD 0.3 million to USD 551 million, and property damage costs from USD 8 million to USD 10 billion. The overall costs of residential fires varied between USD 13 million and USD 435 million.

Four of the selected studies [18,42–44] were conducted in the USA. Lawrence et al. [44] and Ahrens and Evarts [18] conducted their studies on the overall population in the USA. The study populations reported in McLoughlin and McGuire [42] and Yellman et al. [43] were not a true representative of the overall population, as McLoughlin and McGuire [42] conducted their study on children up to 19 years old, and Yellman et al. [43] reported on the recipient and non-recipient households participating in the smoke alarm installation program in Dallas. Only Yellman et al. [43] described the number of participants in the installation program; none of the other three studies [18,42,44] reported the population figures in their studies. The reporting criteria of the studies were very diverse. McLoughlin and McGuire [42] estimated the costs of fire and burn deaths in children and modelled the societal costs of these deaths by measuring the loss of life in years and the lost productivity that imposed enormous economic burdens on families and society. Yellman et al. [43] modelled the economic effectiveness of smoke alarm installation and concluded that when estimates of the lost potential labour productivity of injured individuals are included, the smoke alarm installation program has a net saving to the society. The focus of Lawrence et al. [44] was on estimating the costs of medically treated injuries, and the loss of productivity and quality of life associated with residential fires through different health services (emergency department, and hospital inpatient and outpatient care). Ahrens and Evarts [18] reported only the cost of property damage related to residential fires along with residential fire incidence and prevalence in the USA. None of the studies reported any costs of response agencies, other business agents, and cost of government to the society.

This review found one study in each of Canada [45], Australia [46], and Kuwait [47], with very different study populations and reporting criteria. In Canada, Banfield et al. [45] conducted a cost analysis of patients aged 16 years and above with burn or inhalation injuries caused by residential fires in Ontario. Tannous et al. [46] conducted their study on a piloted program, Home Fire Safety Checks (HFSC), in NSW, Australia. Koushki and Terro [47] conducted their study on a research project aiming to determine the general characteristics of building fires and to develop models to estimate the costs of fires and associated injuries in Kuwait. None of them described the population figures in their studies. Banfield et al. [45] reported the health service costs including the costs of treatment, ambulance service, and hospital inpatient/outpatient care. They further reported the loss of property for the reported residential fires. They did not assess the lost productivity and quality of life, costs of response agencies, other business agents, and cost of government to the society. In their study, Tannous et al. [46] used estimated costing figures from other published studies on costings while quantifying the health system costs, loss of statistical value of life and quality of life in the form of measuring the pain and suffering of injured people, the costs of response agencies and insurance companies, and the cost to the community associated with residential fires. They did not report the cost of government to the society. Koushki and Terro [47] quantified only the cost of property damage in their study. Though they reported that 69% of all building fires occurred in residential properties, they did not describe any other costs associated with residential fires.

This systematic review did not identify any study in Europe, in New Zealand, in other Asian countries, or in African and Latin American countries that estimated and reported on the costs of residential fires and associated injuries and deaths. There were insufficient studies on the costs to individuals (for example, lost income/productivity

and quality of life), costs of response agencies (such as fire services, ambulance, police, and emergency services), costs of health services (for instance, emergency department, and hospital inpatient/outpatient care), and costs of other businesses (for example, utility providers and insurance companies) to provide a robust cost estimate for the stakeholders. Further, this review found that the costs of government (in terms of the lost tax revenue, the cost of welfare payment, and funding/support to different response agencies and to the society) related to residential fires have not been reported yet. No study captured a total picture of the costs of residential fires from the societal and economic perspective including all stakeholders.

The strength of this systematic review is its search strategy which was comprehensive in nature. It included a wide period of time and any available studies from all countries following the PRISMA guidelines. The quality of the studies was assessed, and all the included studies were high-quality studies. A customised quality assessment tool, combining commonly used tools (CASP, SIGN, NICE, JBI, and Newcastle-Ottawa Scale) was considered. This review did not evaluate any intervention of the included studies, and, therefore, used a customised quality assessment tool for assessing the quality of the included studies. The main limitation of this review is the paucity of reported data and the range of the studies selected. In addition, this review was unable to undertake quantitative analysis to determine a robust economic cost of residential fires in total and/or to specific stakeholder(s).

## 5. Conclusions

This systematic review provides the most comprehensive list of studies on residential fires, specifically, those that included economic costs of these incidents, up to the present date. The studies reported variously on the costs of injuries, deaths, loss of productivity, and quality of life, impacts on the health service system, emergency response agencies, and property damage resulting from residential fires, all measured in both current and constant 2022 US dollars. Notably, none of the seven included studies reported on all these costs. While direct comparability was challenging, the findings offer valuable insights into the diverse and extensive impacts of residential fires, extending beyond the current focus on the number of injuries or deaths. The ripple effect of reduced individual productivity also extends to employer, co-workers, and potential outputs. Additionally, the consequences on the quality of life affect not only the individual(s) but also their families and others in their lives. The findings of this review hold importance for individuals, communities, and government agencies for policy development and planning purposes. The reported costs associated with injuries and deaths may be effectively utilized in fire prevention messaging aimed at influencing individuals' behaviour towards fire safety. Furthermore, this review provides evidence on the impact of residential fires on property damage, which may inform the discussion on the importance of implementing fire mitigation measures for property developers and/or property owners. Insurance companies, in particular, may find this information valuable in their communications with policyholders, both in terms of reducing fire risks and pricing various general insurance products.

Future studies are recommended to investigate the comprehensive economic costs of residential fires from a societal perspective, as this aspect has been identified as inadequately researched and documented. Particularly, no study was found regarding the economic costs of residential fires in LMICs, highlighting a research gap in this area. These types of studies are of paramount importance in the current context, given the rising global temperatures and the increasing frequency of conditions conducive to the ignition and rapid spread of residential fires.

**Author Contributions:** W.K.T. and L.A.H. conceived and designed the systematic review. L.A.H. guided the review at every step. F.S.R. developed the search history and a customised risk of bias tool, searched the databases and grey literature, and sorted studies based on inclusion and exclusion criteria. F.S.R., N.G. and L.A.H. screened the selected studies by title and abstract and then finalised the eligible studies by full text review. F.S.R., W.K.T. and G.A. extracted the data from the selected

eligible studies. F.S.R. analysed the data and wrote the manuscript. W.K.T. and L.A.H. reviewed and edited the manuscript. F.S.R., W.K.T., G.A., K.E.A., N.G. and L.A.H. reviewed and approved the final version of the systematic review and agreed to take any accountability for all aspects of the work. All authors have read and agreed to the published version of the manuscript.

**Funding:** The Fire and Rescue New South Wales (FRNSW), Australia, and Western Sydney University (WSU) in Sydney, Australia, have been providing a 'Western Sydney University and Fire and Rescue NSW Postgraduate Research Scholarship' (Grant No: P00027201) to support Fahmida's PhD study. The Digital Health Cooperative Research Centre (DHCRC) Australia has been providing a 'Top-up scholarship' (Grant No: P00028109) to support Fahmida's PhD study.

**Institutional Review Board Statement:** Not required.

**Informed Consent Statement:** This study does not involve human participants and animal subjects. It is a systematic review, data were collected from published material, and thereby ethics approval was not needed. Meanwhile, ethical approval for the PhD study has been provided by the NSW Population and Health Service Research Ethics Committee (HREC/16/CIPHS/36) and Western Sydney University Human Research Ethics Committee (RH12399).

**Data Availability Statement:** Data was collected from published material.

**Acknowledgments:** We are thankful to the Graduate Research School (GRS) in Western Sydney University for providing all supports to Fahmida's PhD journey. We would like to express our gratitude to Paul Jewell (Librarian, School of Business, Western Sydney University) and Lily Collison (Librarian, School of Medicine, Western Sydney University) for their help in developing search terms and guidance during the initial search process.

**Conflicts of Interest:** The authors declare no conflict of interest.

## Appendix A

**Table A1.** Search history on databases.

| Search term | Database: MEDLINE Results |
|---|---|
| 1. residen* or home* or hous* or building* or apartment* or kitchen* or bedroom* or living room* or laundr* or dryer* or structur* or office* or domestic* or electr* appliance* | 3,429,307 |
| 2. fire* or smoke* or flame* | 206,710 |
| 3. injur* or burn* or wound* or damage* or harm* or death* or mortal* or fatal* or casualt* | 3,208,238 |
| 4. cost* or burden* or damage* or individual* or bystander* or carer* or product* or income* or outcome* or working hour* or business* or industr* or agenc* or government* or insurance or utility or maintenance or hospital* or ambulance* or fire brigade* or police* or propert* or structur* or societal or economic or health or treatment or false alarm* | 934,344 |
| 1 AND 2 AND 3 AND 4 | 934 |
| Limit to English language and year: 1978–2020 | 878 |
| Search term | Database: EMBASE Results |
| 1. residen* or home* or hous* or building* or apartment* or kitchen* or bedroom* or living room* or laundr* or dryer* or structur* or office* or domestic* or electr* appliance* | 4,144,589 |
| 2. fire* or smoke* or flame* | 317,348 |
| 3. injur* or burn* or wound* or damage* or harm* or death* or mortal* or fatal* or casualt* | 4,814,620 |
| 4. cost* or burden* or damage* or individual* or bystander* or carer* or product* or income* or outcome* or working hour* or business* or industr* or agenc* or government* or insurance or utility or maintenance or hospital* or ambulance* or fire brigade* or police* or propert* or structur* or societal or economic or health or treatment or false alarm* | 1,434,344 |
| 1 AND 2 AND 3 AND 4 | 1496 |
| Limit to English language and year: 1978–2020 | 1394 |



**Table A1.** *Cont.*

| | Database: ECONLIT |
|---|---|
| Search term | Results |
| 1. residen* or home* or hous* or building* or apartment* or kitchen* or bedroom* or living room* or laundr* or dryer* or structur* or office* or domestic* or electr* appliance* | 297,436 |
| 2. fire* or smoke* or flame* | 4262 |
| 3. injur* or burn* or wound* or damage* or harm* or death* or mortal* or fatal* or casualt* | 37,445 |
| 4. cost* or burden* or damage* or individual* or bystander* or carer* or product* or income* or outcome* or working hour* or business* or industr* or agenc* or government* or insurance or utility or maintenance or hospital* or ambulance* or fire brigade* or police* or propert* or structur* or societal or economic or health or treatment or false alarm* | 168,886 |
| 1 AND 2 AND 3 AND 4 | 49 |
| Limit to English language and year: 1978–2020 | 49 |
| | Database: CINAHL |
| Search term | Results |
| 1. residen* or home* or hous* or building* or apartment* or kitchen* or bedroom* or living room* or laundr* or dryer* or structur* or office* or domestic* or electr* appliance* | 614,181 |
| 2. fire* or smoke* or flame* | 63,075 |
| 3. injur* or burn* or wound* or damage* or harm* or death* or mortal* or fatal* or casualt* | 719,271 |
| 4. cost* or burden* or damage* or individual* or bystander* or carer* or product* or income* or outcome* or working hour* or business* or industr* or agenc* or government* or insurance or utility or maintenance or hospital* or ambulance* or fire brigade* or police* or propert* or structur* or societal or economic or health or treatment or false alarm* | 260,672 |
| 1 AND 2 AND 3 AND 4 | 340 |
| Limit to English language and year: 1978–2020 | 332 |
| | Database: SCOPUS |
| Search term | Results |
| 1. residen* or home* or hous* or building* or apartment* or kitchen* or bedroom* or living room* or laundr* or dryer* or structur* or office* or domestic* or electr* appliance* | 11,609,676 |
| 2. fire* or smoke* or flame* | 652,070 |
| 3. injur* or burn* or wound* or damage* or harm* or death* or mortal* or fatal* or casualt* | 5,207,383 |
| 4. cost* or burden* or damage* or individual* or bystander* or carer* or product* or income* or outcome* or working hour* or business* or industr* or agenc* or government* or insurance or utility or maintenance or hospital* or ambulance* or fire brigade* or police* or propert* or structur* or societal or economic or health or treatment or false alarm* | 3,109,074 |
| 1 AND 2 AND 3 AND 4 | 4174 |
| Limit to English language and year: 1978–2020 | 3758 |

**Table A2.** Quality assessment and risk of bias tool used in the systematic review.

| Domain | Potential Question | CASP | SIGN | NICE | JBI | NOS |
|---|---|---|---|---|---|---|
| Selection | Is the source population or source area well described? | | | x | | |
| Exposure | Was the exposure accurately measured to minimise bias? | x | x | x | x | x |
| Outcome assessment | Were the outcomes clearly defined? | | x | x | | |
| Selective reporting | Were all measured outcomes reported? | | | | | |
| Generalisability | Can the outcomes be applied to the general population? | x | x | | | |

**Table A3.** List of full-text studies and reports excluded for failing to meet the eligibility criteria.

| Articles | Number of Articles | Reasons for Exclusion |
|---|---|---|
| Woodward [52]; Wood [53]; Weber and Smith [54]; Harvey, Ghassempour [26]; Stringfellow [55]; Fire and Rescue New South Wales [56]; Rural Fire Service New South Wales [57]; Burns Registry of Australia and New Zealand [58]; Ashe, McAneney [24]; World Health Organization [2]; World Health Organization [1]; Tannous and Agho [27]; Mattsson and Juås [59]; Leistikow, Martin [60]; Silverstein and Lack [61]; Zhuang, Payyappalli [62] | 16 | Does not include relevant data |
| Salter, Ramachandran [63]; Vong [64]; Eckler [65] | 3 | Non-residential |
| Shields, Gielen [66]; Rehou, Banfield [67] | 2 | Abstract |
| Salka Jr [68]; Siarnicki [69] | 2 | Editorial |
| Campanharo, Lopes [70] | 1 | About bush fire |
| Kazerooni, Gyedu [71] | 1 | Systematic review |
| Beaulieu, Smith [72]; Stephen G. Badger [73] | 2 | Data covered by other selected studies |
| Total | 27 | |

**Table A4.** List of studies and reports excluded for not reporting on costs of residential fires.

| Author and Year of Publication | Country | Study Population | Extrapolated Population (for the Purpose of Analysis) | Study Design | Data Collection Period | Type of Data | Number of Residential Fires, Associated Injuries and Deaths | Type of Publication |
|---|---|---|---|---|---|---|---|---|
| Istre, 2001 [74] | Dallas, USA | 1,006,877 in Dallas in 1990 | - | Observational | 1991–1997 | Administrative data | Fires: 1717 annually; injuries: 3.4 per 100,000 population annually; deaths: 1.8 per 100,000 population annually | Peer-reviewed journal |
| Istre, 2014 [75] | Dallas, USA | 107,705 residents participating in a community-based smoke alarm installation program | - | Observational | 2001–2011 | Administrative and survey data | Injuries: 1.5 per 100,000 population annually; deaths: 2.4 per 100,000 population annually | Peer-reviewed journal |
| System Planning Corporation, 2009 [4] | Ontario, Canada | At the population level | 12,883,583 [76] in Ontario in 2008 | Observational | 2003–2004 | Administrative and survey data | Deaths: 97 [0.8 per 100,000 population] annually | Commissioned report |
| System Planning Corporation, 2009 [4] | Ottawa, Canada | At the population level | 1,176,000 [77] in Ottawa in 2008 | Observational | 2003–2004 | Administrative and survey data | Deaths: 0.34 per 100,000 population annually | Commissioned report |
| System Planning Corporation, 2009 [4] | Toronto, Canada | At the population level | 5,309,000 [78] in Toronto in 2008 | Observational | 2003–2004 | Administrative and survey data | Deaths: 0.7 per 100,000 population annually | Commissioned report |

**Table A4.** *Cont.*

| Author and Year of Publication | Country | Study Population | Extrapolated Population (for the Purpose of Analysis) | Study Design | Data Collection Period | Type of Data | Number of Residential Fires, Associated Injuries and Deaths | Type of Publication |
|---|---|---|---|---|---|---|---|---|
| System Planning Corporation, 2009 [4] | Vancouver, Canada | At the population level | 2,198,000 [79] in Vancouver in 2008 | Observational | 2003–2004 | Administrative and survey data | Deaths: 0.5 per 100,000 population annually | Commissioned report |
| Beaulieu, 2020 [19] | British Columbia, Alberta, Manitoba, and Ontario, Canada | At the population level in British Columbia, Alberta, Manitoba, and Ontario, Canada | 22,572,492 [80] on average annually in those four provinces during 2005–2015 | Observational | 2005–2015 | Administrative data | Fires: 13,205 annually; injuries: 852 [3.8 per 100,000 population] annually; deaths: 134 [0.6 per 100,000 population] annually | Peer-reviewed journal |
| System Planning Corporation, 2009 [4] | Puerto Rico | At the population level | 3,725,595 [81] on average annually during 1994–2007 | Observational | 2003–2004 | Administrative and survey data | Deaths: 16 [0.4 per 100,000 population] annually | Commissioned report |
| System Planning Corporation, 2009 [4] | Santo Domingo | At the population level | 2,457,000 [82] in 2008 | Observational | 2003–2004 | Administrative and survey data | Deaths: 4 [0.2 per 100,000 population] annually | Commissioned report |
| System Planning Corporation, 2007 [83] | UK | At the population level | 60,383,741 [84] in 2005 | Observational | 2003–2004 | Administrative and survey data | Deaths: 342 [0.6 per 100,000 population] annually | Commissioned report |
| Holborn, 2003 [85] | London, England | At the population level and reported on fatal fire incidences | 7,175,500 [86] on average annually during 1996–2000 | Observational | 1996–2000 | Administrative data | Fires: 64 annually; deaths: 72 [1.0 per 100,000 population] annually | Peer-reviewed journal |
| System Planning Corporation, 2007 [83] | England | At the population level | 50,606,000 [87] in 2005 | 2005 | 2003–2004 | Administrative and survey data | Deaths: 259 [0.5 per 100,000 population] annually | Commissioned report |
| Beasley, 2018 [88] | London, England | At the population level and reported on residential fires caused by household appliances | 8,417,500 [89] on average annually during 2011–2015 | Observational | 2011–2015 | Administrative data | Fires: 206 annually; injuries: 33 [0.4 per 100,000 population] annually; deaths: 1.6 [0.02 per 100,000 population] annually | Peer-reviewed journal |

**Table A4.** *Cont.*

| Author and Year of Publication | Country | Study Population | Extrapolated Population (for the Purpose of Analysis) | Study Design | Data Collection Period | Type of Data | Number of Residential Fires, Associated Injuries and Deaths | Type of Publication |
|---|---|---|---|---|---|---|---|---|
| Fire and Rescue, 2021 [20] | England | At the population level | 56,489,800 [90] in 2021 | Observational | 2020–2021 | Administrative data | Fires: 27,121; injuries: 4877 [8.6 per 100,000 population]; deaths: 185 [0.3 per 100,000 population] | Annual report |
| Fire and Rescue, 2021 [91] | Wales | At the population level | 3,107,500 [92] | Observational | 2020–2021 | Administrative data | Fires: 1501; injuries: 332 [10.7 per 100,000 population]; deaths: 19 [0.6 per 100,000 population] | Annual report |
| System Planning Corporation, 2007 [83] | Scotland | At the population level | 5,110,200 [93] in 2005 | Observational | 2003–2004 | Administrative and survey data | Deaths: 56 [1.1 per 100,000 population] annually | Commissioned report |
| Fire and Rescue (2020) [94] | Scotland | At the population level and reported on accidental residential fires | 5,479,900 [95] in 2020 | Observational | 2019–2020 | Administrative data | Fires: 4339; injuries: 573 [10.5 per 100,000 population]; deaths: 21 [0.4 per 100,000 population] | Annual report |
| Fire and Rescue, 2021 [96] | Northern Ireland | At the population level and reported on accidental residential fires | 1,903,100 [97] in 2021 | Observational | 2020–2021 | Administrative data | Fires: 761; injuries: 83 [4.4 per 100,000 population]; deaths: 8 [0.4 per 100,000 population] | Annual report |
| System Planning Corporation, 2007 [83] | Sweden | At the population level | 9,164,272 [98] in 2007 | Observational | 20003–2004 | Administrative and survey data | Deaths: 860 [9.4 per 100,000 population] annually | Commissioned report |
| Jonsson, 2017 [23] | Sweden | At the population level and reported on fatal fire incidences | 9,022,177 [99] on average annually during 1999–2007 | Observational | 1999–2007 | Administrative and secondary data | Fires: 92 annually; deaths: 99 [1.1 per 100,000 population] annually | Peer-reviewed journal |
| Sund, 2019 [100] | Southern Sweden | 520,000 | - | Observational | 2000–2015 | Administrative data | Fires: 316 annually; deaths: 7 [1.3 per 100,000 population] annually | Peer-reviewed journal |

**Table A4.** *Cont.*

| Author and Year of Publication | Country | Study Population | Extrapolated Population (for the Purpose of Analysis) | Study Design | Data Collection Period | Type of Data | Number of Residential Fires, Associated Injuries and Deaths | Type of Publication |
|---|---|---|---|---|---|---|---|---|
| System Planning Corporation, 2007 [83] | Norway | At the population level | 4.661,087 [101] in 2006 | Observational | 2003–2004 | Administrative and survey data | Deaths: 62 [1.3 per 100,000 population] annually | Commissioned report |
| Sesseng, 2017 [102] | Norway | At the population level and reported on fatal fires incidences | 9,761,612 [103] on average annually during 2005–2014 | Observational | 2005–2014 | Administrative data | Deaths: 46 [0.5 per 100,000 population] annually | Commissioned report |
| System Planning Corporation, 2008 [104] | Australia | At the population level | 20,467,030 [105] in 2006 | Observational | 2003–2004 | Administrative and survey data | Deaths: 70 [0.3 per 100,000 population] annually | Commissioned report |
| Fire and Rescue NSW, 2020 [106] | NSW, Australia | At the population level and reported on accidental residential fires | - | Observational | 2019–2020 | Administrative data | Fires: 93.2 per 100,000 people | Annual report |
| Fire and Rescue NSW, 2015 [107] | NSW, Australia | At the population level and reported on the winter season | 23,820,236 [108] in 2015 | Observational | 2014–2015 | Administrative data | Fires: 4254 | Annual report |
| Steering Committee for the Review of Government Service Provision, 2021 [25] | Australia | At the population level and reported on accidental residential fires | - | Observational | 2020–2021 | Administrative data | Fires: 75.1 per 100,000 households | Annual report |
| Coates, 2019 [5] | Australia | At the population level | 22,144,667 [109] on average annually during 2003–2017 | Observational | 2003–2017 | Administrative data | Deaths: 64 [0.3 per 100,000 population] annually | Organisational report |
| Xiong, 2015 [110] | NSW, VIC, and QLD, Australia | Reported on adult (18 years and above) population | 16,755,700 [111] in the three states in 2008 | Observational | 1998–2008 | Administrative data | Deaths: 20 [0.1 per 100,000 population] annually | Peer-reviewed journal |
| System Planning Corporation, 2008 [104] | New Zealand | At the population level | 4,179,978 [112] in 2006 | Observational | 2003–2004 | Administrative and survey data | Deaths: 0.35 per 100,000 population | Commissioned report |
| Fire and Emergency, 2020 [113] | New Zealand | At the population level and reported on avoidable fatalities | 5,090,200 [114] in 2020 | Observational | 2019–2020 | Administrative data | Deaths: 11 [0.2 per 100,000 population] | Annual report |

**Table A4.** *Cont.*

| Author and Year of Publication | Country | Study Population | Extrapolated Population (for the Purpose of Analysis) | Study Design | Data Collection Period | Type of Data | Number of Residential Fires, Associated Injuries and Deaths | Type of Publication |
|---|---|---|---|---|---|---|---|---|
| Fire and Emergency, 2018 [115] | New Zealand | At the population level and reported on houses with sprinklers | 4,398,550 [116] on average annually during 2000–2018 | Observational | 2000–2018 | Administrative data | Fires: 3483 annually; injuries: 1.5 [0.03 per 100,000 population] annually; deaths: 2 in total [0.0 per 100,000 population] | Organisational report |
| Fire and Emergency, 2018 [117] | Auckland, New Zealand | At the population level | 1,407,000 [118] on average annually during 2007–2017 | Observational | 2007–2017 | Administrative data | Fires: 615 annually; injuries: 49 [3.5 per 100,000 population] annually; deaths: 3 [0.2 per 100,000 population] annually | Organisational report |
| Waller, 1998 [119] | New Zealand | 2.56 million in 1988, adults 15 years and above | - Reported fatal residential fires and thermal injury deaths | Observational | 1978–1987 | Administrative data | Fires: 22 annually; injuries: 0.5 per 100,000 population in 1988; deaths: 22 [0.9 per 100,000 population] annually | Peer-reviewed journal |
| Shao, 2012 [120] | Taiwan | At the population level and reported on 17 fire incidents | 21,166,167 [121] on average annually during 1984–2010 | Observational | 1984–2010 | Administrative data | Fires: 2346 annually; injuries: 164 [0.8 per 100,000 population]; deaths: 206 [1.0 per 100,000 population] over the time | Peer-reviewed journal |
| Sekizawa, 2006 [122] | Japan | At the population level | - | Observational | 1983–1987 | Administrative data | Deaths: 21.3 fire deaths per year per million units of houses | Peer-reviewed journal |
| System Planning Corporation, 2008 [104] | Japan | At the population level | 128,006,426 [123] in 2007 | Observational | 2003–2004 | Administrative and survey data | Deaths: 1357 [1.1 per 100,000 population] | Commissioned report |

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
