# Peer review of "Economic Costs of Residential Fires: A Systematic Review"

_fire, doi:10.3390/fire6100399_

Round 1

Reviewer 1 Report

The paper concerns the problem of the estimation of economic cost of residential fires. The problem is actual and relevant, and the paper concerning this problem may contribute to the domain. The research plan was very well designed and conducted. I have no any concern to this part of the research. The research process was also very well described and documented in the paper.
My only concern is related to the discussion and conclusion. They are limited only to discuss what other authors tried to do in their research. However, I would expect, summary of what has been found in the articles. Even with the low credibility there should be attempt to estimate the cost of residential fires from various perspectives and various stakeholders. A literature review without conclusions is not the whole product.  

Very well written. 

Author Response

Please find enclosed the response to reviewer 1.

Reviewer 2 Report

The manuscript ID: fire-2622678 focused on the review of economic costs of residential fires. This study is well structured and follows a strict methodology design. Although it is an interesting research topic and the study contains some valuable information, some issues need to be clarified before further processing.

1. The authors should make the research question more clear. It is still not very clear what key question this study is to answer or what problem this study is to solve. It may not be enough to document the extensive literature and only to report on statistics of economic costs of residential fires.

2. It may be inappropriate to only keep 7 studies in "a systematic review" out of thousands of potential literatures. The sample size of 7 is too small. The results are thus sensitive to the cases selected in the sample and the degree of generalisability is weakened. For example, the sample studies all focus on high-income countries, with no sample on low and medium-income countries. The authors should think about the eligibility criteria and make this research more comprehensive and more useful for decision-making.

3. The expressions need to be more rigorous. For example, in conclusion section, "Of the identified studies, the majority were in the grey literature". The reality is, as section 3.1 illustrated, the electronic database searches yielded a total of 7,020 records, while only 33 records identified by hand searching from grey literature.

Author Response

Please find enclosed the response to reviewer 2.

Reviewer 3 Report

Dear Authors,

The manuscript reviewed the economic expenses of residential fires from a societal perspective comprehensively. Such a manuscript can help the society to estimate fire’s harms leading to paying much attention to prevention of fire.  Generally, I appreciate the study because it is so absorbing that I read it two times and it has been written accurately. I believe the manuscript can be published in its current form. Briefly, Congratulation for such a study.

Regards

Author Response

Please find enclosed the response to reviewer 3.

Round 2

Reviewer 1 Report

The authors answered my concerns and updated the paper. Although the conclusions still do not contain quantitative summary of the survey, there is explanation why at the current moment it is impossible. I think that in this form the paper can be published. 

Minor corrections are needed. 

Reviewer 2 Report

The manuscript is OK.